# Quantitation of the Surface Shortwave and Longwave Radiative Effect of Dust with an Integrated System: A Case Study at Xianghe

**DOI:** 10.3390/s24020397

**Published:** 2024-01-09

**Authors:** Mengqi Liu, Hongrong Shi, Jingjing Song, Disong Fu

**Affiliations:** 1Key Laboratory of Atmospheric Sounding, Chengdu University of Information Technology, Chengdu 610225, China; 2Key Laboratory of Middle Atmosphere and Global Environment Observation (LAGEO), Institute of Atmospheric Physics, Chinese Academy of Sciences, Beijing 100029, China; shihrong@mail.iap.ac.cn; 3Beijing Meteorological Data Center, Beijing 100097, China

**Keywords:** dust, radiative effect, North China Plain

## Abstract

Aerosols play a crucial role in the surface radiative budget by absorbing and scattering both shortwave and longwave radiation. While most aerosol types exhibit a relatively minor longwave radiative forcing when compared to their shortwave counterparts, dust aerosols stand out for their substantial longwave radiative forcing. In this study, radiometers, a sun photometer, a microwave radiometer and the parameterization scheme for clear-sky radiation estimation were integrated to investigate the radiative properties of aerosols. During an event in Xianghe, North China Plain, from 25 April to 27 April 2018, both the composition (anthropogenic aerosol and dust) and the aerosol optical depth (AOD, ranging from 0.3 to 1.5) changed considerably. A notable shortwave aerosol radiative effect (SARE) was revealed by the integrated system (reaching its peak at −131.27 W·m^−2^ on 26 April 2018), which was primarily attributed to a reduction in direct irradiance caused by anthropogenic aerosols. The SARE became relatively consistent over the three days as the AODs approached similar levels. Conversely, the longwave aerosol radiative effect (LARE) on the dust days ranged from 8.94 to 32.93 W·m^−2^, significantly surpassing the values measured during the days of anthropogenic aerosol pollution, which ranged from 0.35 to 28.67 W·m^−2^, despite lower AOD values. The LARE increased with a higher AOD and a lower Ångström exponent (AE), with a lower AE having a more pronounced impact on the LARE than a higher AOD. It was estimated that, on a daily basis, the LARE will offset approximately 25% of the SARE during dust events and during periods of heavy anthropogenic pollution.

## 1. Introduction

Atmospheric aerosols have an essential influence on Earth’s radiative budget, contributing to various effects on both shortwave and longwave radiation, which are characterized by large uncertainties. Aerosols perturb the energy budget of the Earth system by absorbing and scattering shortwave radiation, as well as by absorbing and reemitting longwave radiation [1]. It is accurate to assert that, for most aerosol types, particularly, fine particles like pollution and smoke, radiative forcing in the longwave range is relatively small compared to their forcing in the shortwave range. Consequently, the longwave aerosol radiative effect (LARE) is commonly neglected [2,3]. However, it is crucial to acknowledge that the LARE becomes nonnegligible when considering highly absorbing and scattering particles, such as mineral dust [4,5]. Dust aerosols are considered a major source of tropospheric aerosol loading and constitute a notable source of uncertainty in climate change modelling [6].

Deserts are the primary source of dust worldwide and thus a focal point for radiative forcing studies. These investigations have predominantly centered on deserts and their surrounding areas, including the Sahara desert, the Mediterranean Sea, the Taklimakan desert, etc. [7,8,9]. The daily shortwave aerosol radiative effect (SARE) of dust varies between −209 W·m^−2^ and −0.5 W·m^−2^, while the daily LARE fluctuates between +2.8 W·m^−2^ and +41.5 W·m^−2^ [4,7,8,10]. Remarkably, the LARE accounts for up to 53% of the SARE [8,11,12]. Presently, a significant level of uncertainty remains in the assessment of dust SARE and LARE, mainly due to uncertainties in the optical and microphysical properties of dust, resulting from the high spatial and temporal variability of its particle size, chemical composition, shape and vertical distribution, all of which affect their light absorption and scattering properties.

The North China Plain (NCP) experiences consistent regional haze all year round, along with occasional large spreads of dust in spring, primarily originating from Inner Mongolia and the Hebei Province. Different from deserts and their surrounding areas, dust aerosols in the NCP exhibit a more complex composition, influenced by a mixture of anthropogenic and natural aerosols. However, there has been limited empirical assessment of the SARE and LARE in this area.

In this study, an integrated system operated at Xianghe, NCP, was utilized to investigate the SARE and LARE (more detailed information is presented in Section 2). A dust event on a day close to polluted, clear-sky days was chosen. The primary aim of this research was twofold: firstly, to estimate the SARE and LARE for a dust event observed at Xianghe in the NCP; and secondly, to conduct a comparative analysis of radiative forcing among different aerosol types and loadings.

The rest of this paper is organized as follows. The data and clean-sky radiation estimation method are described in Section 2; the results are discussed in Section 3; and the conclusions are drawn in Section 4.

## 2. Data and Method

An overview of the integrated system is shown in Figure 1. Specifically, radiation was measured with a pyranometer, a pyranometer with a shadow ball, and a pyrgeometer; the aerosol properties were collected by a sun photometer; meteorological parameters were obtained by a microwave radiometer. All the measurements mentioned above subsequently served as inputs in the parameterization scheme for clear-sky radiation calculation.

### 2.1. Site and Measurements

The instruments used in this study were installed at the Xianghe Integrated Observatory (39°45′ N, 116°57′ E, and 35 m a.s.l.). Xianghe is located in the NCP, approximately 60 km southeast of Beijing. The main sources of aerosols in this area are anthropogenic, from domestic industry and traffic, and dust during the spring season. Global horizontal irradiance (GHI), direct normal irradiance (DNI), diffuse horizontal irradiance (DHI), downward longwave radiation (DLR), etc., have been measured since 2004. Additionally, continuous monitoring of aerosol and meteorological parameters has been ongoing by a sun photometer and a microwave radiometer.

At this observatory, GHI measurements are obtained using a Kipp&Zonen CMP21 pyranometer (Delft, The Netherlands) (only since 2013), while DNI and DHI are measured separately with a CHP1 pyrheliometer and a Kipp&Zonen CMP21 with a shadow ball. For DLR, a Kipp&Zonen CGR4 pyrgeometer is used in the spectral range of 450–4200 nm range. The radiometers undergo calibration every 2 years to maintain accuracy. A quality control (QC) process was performed on the shortwave and longwave radiation data following the procedures outlined by the Baseline Surface Radiation Network (BSRN) [13].

Aerosol optical depth (AOD), Ångström exponent (AE) and precipitable water vapor (WV) are obtained by a CIMEL CE-318 sun photometer (Voutré, France), which is part of the AERONET network [14]. This sun photometer conducts direct solar measurements at 340, 380, 440, 500, 675, 870, 1020 and 1640 nm. CE-318 has been operated and calibrated annually at Xianghe since September 2004 [15].

Screen-level temperature (T) and absolute humidity (AH) data are measured by a RPG-HATPRO-G3 (Leeds, UK), a high-precision microwave radiometer for continuous atmospheric profiling, while water vapor pressure (e) was calculated using the T and AH values. The RPG-HATPRO has been operating at Xianghe since March 2012 and is regularly calibrated (1–2 times/year) at the site. In addition, to address the gaps in the data because of missing values, the T and e values measured at Tongzhou (39°50′ N, 116°45′ E), a neighboring site of Xianghe, were used in this study.

### 2.2. Event Selection

Spring dust events in Xianghe, identified based on aerosol properties (AOD and AE), were carefully analyzed. These dust events were defined by the simultaneous fulfillment of AOD > 0.4 and AE < 0.6 conditions. From the extensive series of dust events spanning from 2015 to 2019, one particular date, namely, 27 April 2018, was chosen for further analysis. The date was characterized by a full day clear-sky condition (visually inspected by the diurnal variation in GHI), while a dust event was recorded.

In addition, to investigate the difference between SARE and LARE for different aerosol types, polluted, clear-sky days (25 and 26 April 2018) close to the chosen dust event (27 April 2018) were also considered. As shown in Figure 2, differences in aerosol characteristics for these 3 days were evident in satellite images from the Moderate Resolution Imaging Spectroradiometer (MODIS, original images from https://wvs.earthdata.nasa.gov/ (accessed on 29 September 2023)). The final dataset used in this study is presented and plotted in Table 1 and Figure 3.

### 2.3. Methodology

The aerosol radiative effect is defined as the difference between the measured flux at the surface and the corresponding flux in the absence of aerosol in the atmosphere under clear-sky conditions. The SARE and LARE at the surface are defined as follows:(1)$$\mathrm{SARE}=\mathrm{GHI}-{\mathrm{GHI}}_{\mathrm{cs}}$$
(2)$$\mathrm{LARE}=\mathrm{DLR}-{\mathrm{DLR}}_{\mathrm{cs}}$$
where GHI_cs_ and DLR_cs_ are the estimated clear-sky GHI and DLR without aerosol.

In fact, the complexity of attenuation mechanisms in the atmosphere and the dynamics changes of atmospheric components pose significant challenges. A number of model simulations and statistical methods have been proposed for estimating clear-sky radiation [16,17,18]. These models, often involving partial models and parameterizations, represent a complex and relatively accurate scheme that takes into account the role of Rayleigh scattering, uniformly mixed gases, water vapor, aerosols, ozone, etc. However, it should be noted that many parameters used in these methods are often difficult to obtain and can introduce additional sources of error in the GHI_cs_ and DLR_cs_ estimates. Consequently, researchers have extensively evaluated various parameterization schemes for GHI_cs_ and DLR_cs_ [19,20,21,22]. Considering the specific observations obtained at Xianghe, the parameterization schemes proposed by [23] and [24], as shown in Equations (3) and (4), were used to estimate GHI_cs_ and DLR_cs_ in this study.
(3)$${\mathrm{GHI}}_{\mathrm{cs}}=\frac{0.9662S_{0}\cos^{2}\mathrm{SZA}f_{d}}{1.085\cos\mathrm{SZA}+0.0022e\left(2.7+\cos\mathrm{SZA}\right)+0.1}$$
(4)$${\mathrm{DLR}}_{\mathrm{cs}}=\left(0.23+0.443{\left(\frac{e}{T}\right)}^{0.13}\right)\sigma T^{4}$$
where *S*_0_ is the solar constant, *f_d_* is the Earth’s orbital eccentricity, *T* is the screen-level temperature, and σ = 5.67 × 10^−8^ W·m^−2^·K^−4^ denotes the Stefan–Boltzmann constant. Two statistical indicators, including the mean absolute error (MAE) and the root-mean-square error (RMSE), defined by Equations (5) and (6), were utilized to evaluate the performance of the models.
(5)$$\mathrm{MAE}=\frac{1}{n}\sum_{i=1}^{n}\left|y_{i}-x_{i}\right|$$
(6)$$\mathrm{RMSE}=\sqrt{\frac{1}{n}\sum_{i=1}^{n}{\left(y_{i}-x_{i}\right)}^{2}}$$

An intercomparison between the measured and the estimated GHI and DLR using MAE and RMSE for a clean- and clear-sky day, 5 May 2018, with AOD = 0.14 (Figure 4), is shown in Figure 5. The estimated GHI and DLR showed MAEs of 9.83 W·m^−2^ and 5.51 W·m^−2^, with RMSEs of 11.34 W·m^−2^ and 6.27 W·m^−2^. However, it is important to note that we did not use the calculated clear-sky DLR in the early morning (before 7:00 in local standard time) in the following study due to the computational errors.

## 3. Results

The daily averaged SARE and LARE during the event days are listed in Table 2. A noteworthy observation can be drawn from the first two rows in Table 2. As the AOD increased from 0.51 to 0.85, the SARE experienced a more than twofold change, from −27.57 W·m^−2^ to −65.29 W·m^−2^; however, the corresponding increase in LARE was relatively modest, corresponding to about 6 W·m^−2^. Among the three days, it is worth noting that even though the daily averaged AOD on 27 April 2018 (0.5) was even slightly lower than that on 25 April 2018, on 27 April 2018, a high value of SARE (−51.45 W·m^−2^) and the largest value of LARE (18.99 W·m^−2^) were measured. This indicated an additional radiative effect due to dust during this episode. This result aligns with previous studies in Portugal [10] and in a western Indo Gangetic Plain [25], with similar dust AOD (0.5–0.6). It was estimated that, on a daily basis, the LARE will offset about 25% of the SARE in the dust event.

Figure 6 shows the diurnal variations in SARE and LARE for the studied event. The SARE varied from −53.53 to −0.15 W·m^−2^ on 25 April, from −131.27 to −0.35 W·m^−2^ on 26 April and from −95.88 to −0.02 W·m^−2^ on 27 April. Between 11:00 and 14:30 (local standard time, LST), the mean SARE for 26 April remained notably higher, at around −90 W·m^−2^, in contrast to the values of ~30 W·m^−2^ and 50 W·m^−2^ for 25 April and 27 April, respectively. After 15:00, the AODs for all three days became close (0.4–0.6), consequently resulting in similar values of SARE. It is worth highlighting that starting from 9:00, an increase in SARE on 27 April from −90 to −50 W·m^−2^ was observed, despite the increasing cosSZA. This was associated with a reduction in AOD from 0.7 to 0.4.

Figure 6b shows the evolution of the LARE during this period. The LARE varied between 0.35 and 28.67 W·m^−2^ on 25 April, between 5.18 and 26.56 W·m^−2^ on 26 April and between 8.94 and 32.93 W·m^−2^ on 27 April. Interestingly, despite a relatively large difference in the daily averaged AOD of about 0.8, the range of LARE for 25 April and 26 April was quite close. However, the date of 27 April stood out notably with a higher LARE ranging from 5 to 10 W·m^−2^ above the other two cases, especially after 15:00, when the AODs were similar. In other words, the dust had a nonnegligible effect on longwave radiation.

The instantaneous values of SARE and LARE on 27 April at a cosSZA of 0.819 (38.05 W·m^−2^ and 15.57 W·m^−2^) were significantly lower than those reported in [8]. In that study, substantially higher SARE and LARE values, exceeding 209 W·m^−2^ and 41 W·m^−2^, for the same cosSZA, were reported. This disparity can be attributed to the following two aspects. Firstly, ref. [8] used a dust event characterized by a considerably high aerosol loading (AOD > 1.4), which led to larger SARE values. Secondly, the event examined in [8] was dominated by excessive desert dust in combination with regional marine aerosols. This particular aerosol composition resulted in a lower AE (<0.1) and a larger LARE.

The measurements of DNI and DHI provided valuable insights into how the aerosol affected the GHI in the studied event, as shown in Figure 7. At 9:00, the DNI values on 27 April were 728.09, 327.71 and 467.51 W·m^−2^, corresponding to AODs of 0.4, 1.5, and 0.7, respectively, while the DHI values were 160.55, 307.58, and 256.87 W·m^−2^. Around 12:00, the DNI on 25 April and 27 April was about 790 and 690 W·m^−2^, corresponding to a DHI of 220 and 260 W·m^−2^, with the same AOD (0.5) but a different AE (0.9 and 0.5). These values led to nearly equivalent GHI levels of about 930 and 880 W·m^−2^. This phenomenon indicated that the high SARE value of dust compared to that of the anthropogenic aerosol was mainly due to a significant decrease in DNI, albeit partly offset by an increase in DHI. After 15:00, the DNI and DHI on all three days were identical.

The assessment of how the aerosol parameters affected the LARE was studied. In Figure 8, it is clear that the LARE increased with increasing AOD and decreasing AE. Remarkably, the LARE at AOD = 0.44 and 0.4 < AE < 0.6 was almost equal to that at AOD > 1.16 and AE > 1.2, which means that the dust aerosol had a much greater effect on the LARE than the anthropogenic aerosol. Nonetheless, it is essential to acknowledge that due to the limited data available for this event, a quantitative analysis of the relationship between aerosol parameters and LARE was not considered in our study.

## 4. Conclusions

With an integrated system, continuous measurements of aerosol optical properties and surface radiative fluxes were conducted at Xianghe. This monitoring recorded a noteworthy dust event on 27 April 2018, along with 2 days of anthropogenic pollution on 25–26 April 2018. During the event, the AOD at 440 nm ranged from 0.3 to 1.5, and the AE ranged from 0.4 to 1.3. Considering the continuous 3-day clear sky, this was a valuable episode allowing for a comparison of aerosol radiative effects between different aerosol types.

The radiative effect was derived by comparing measured and parameterized clear-sky radiation. The daily averaged SARE experienced a more than twofold decrease from −27.57 W·m^−2^ to −65.29 W·m^−2^ as the AOD increased from 0.51 to 0.85; however, the dust day showed the largest LARE of 20.21 W·m^−2^. It was estimated that, on a daily basis, the LARE will offset about 25% of the SARE in the dust event, which is similar to what observed in the heavy anthropogenic pollution event.

The instantaneous SARE reached −131.27 W·m^−2^ on 26 April, and the LARE reached 32.93 W·m^−2^ on 27 April. Compared with the anthropogenic aerosol, the high SARE value of dust was mainly due to a significant decrease in DNI, although compensated by an increased DHI to some extent. When the AODs on the three days became closer (0.4–0.6) after 15:00, the SAREs showed quite similar values; however, the LARE on the dust day was obviously higher than those measured on the other two anthropogenically polluted days of 5–10 W·m^−2^. It is noteworthy that the LARE increased with the increase in AOD and the decrease in AE, with a low AE exerting a much greater effect on LARE than a high AOD.

Given the constraints of the available data, broader comparisons were not feasible. At present, given the similar atmospheric and aerosol characteristics observed in the North China Plain, we suggest that these results offer a good representation at a regional scale. The aerosol radiative effect depends on aerosol properties, and its response to AOD in the same atmospheric background should be universal. Moreover, there are numerous GHI products based on satellite observations [26,27], as well as derived radiation products such as DNI and DLR obtained through the physical model chain [28]. In the next step, we will utilize these satellite products to assess regional aerosol radiative effects. Therefore, further investigations are imperative to refine our understanding of dust aerosol radiative effects and mitigate the associated uncertainties.

## Figures and Tables

**Figure 1 sensors-24-00397-f001:**
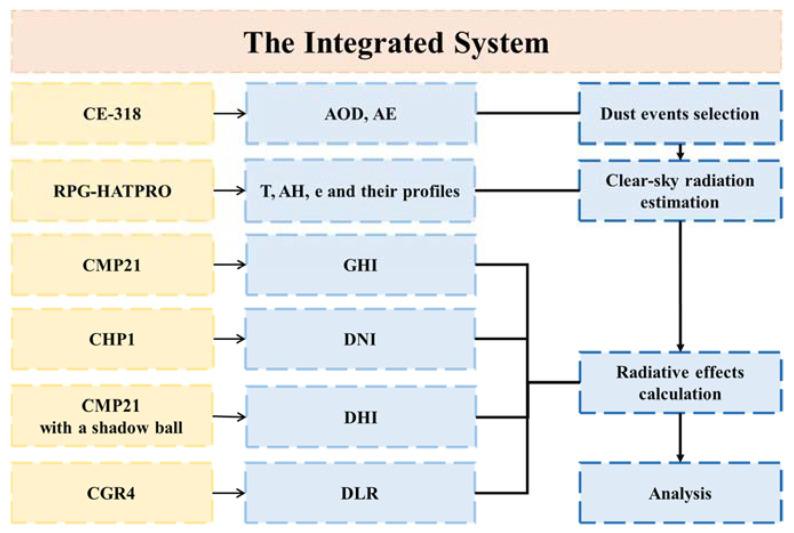
Overview of the integrated system for the quantitation of the SARE and LARE of dust. AOD, AE, T, AH, e, GHI, DNI, DHI and DLR represent aerosol optical depth, Ångström exponent, screen-level temperature, absolute humidity, water vapor pressure, global horizontal irradiance, direct normal irradiance, diffuse horizontal irradiance, downward longwave radiation.

**Figure 2 sensors-24-00397-f002:**
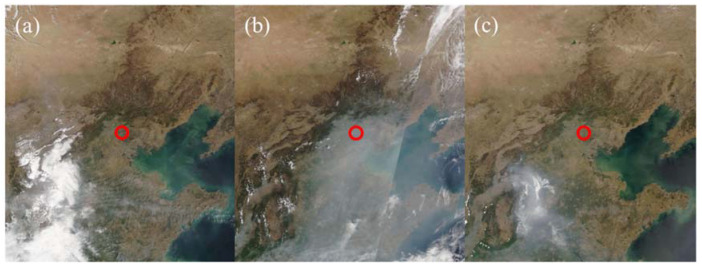
Images of the NCP captured by MODIS on board the Terra satellite for (**a**) 25, (**b**) 26 and (**c**) 27 April 2018. The red circles identify the position of Xianghe.

**Figure 3 sensors-24-00397-f003:**
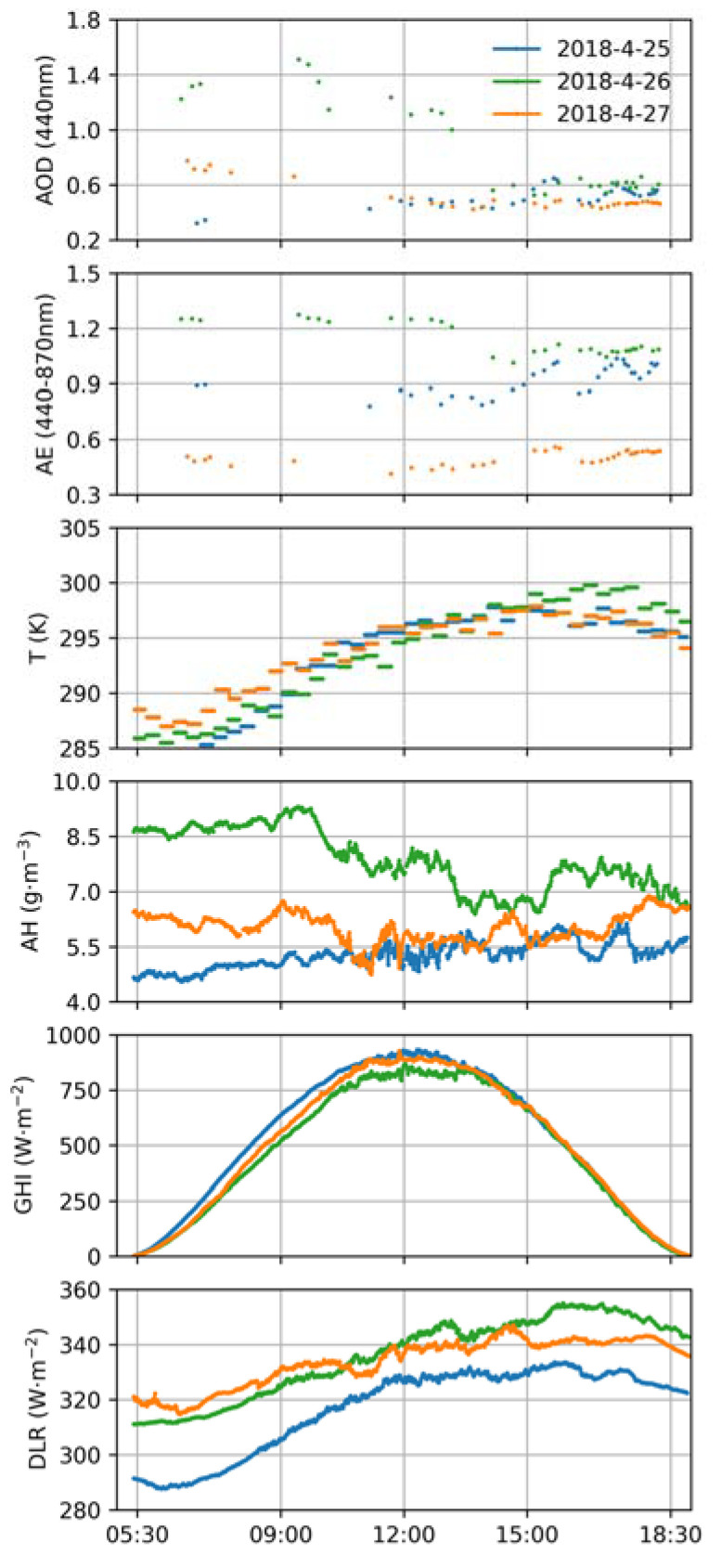
Variation in AOD, AE, T, AH, GHI and DLR on 25–27 April 2018 at Xianghe.

**Figure 4 sensors-24-00397-f004:**
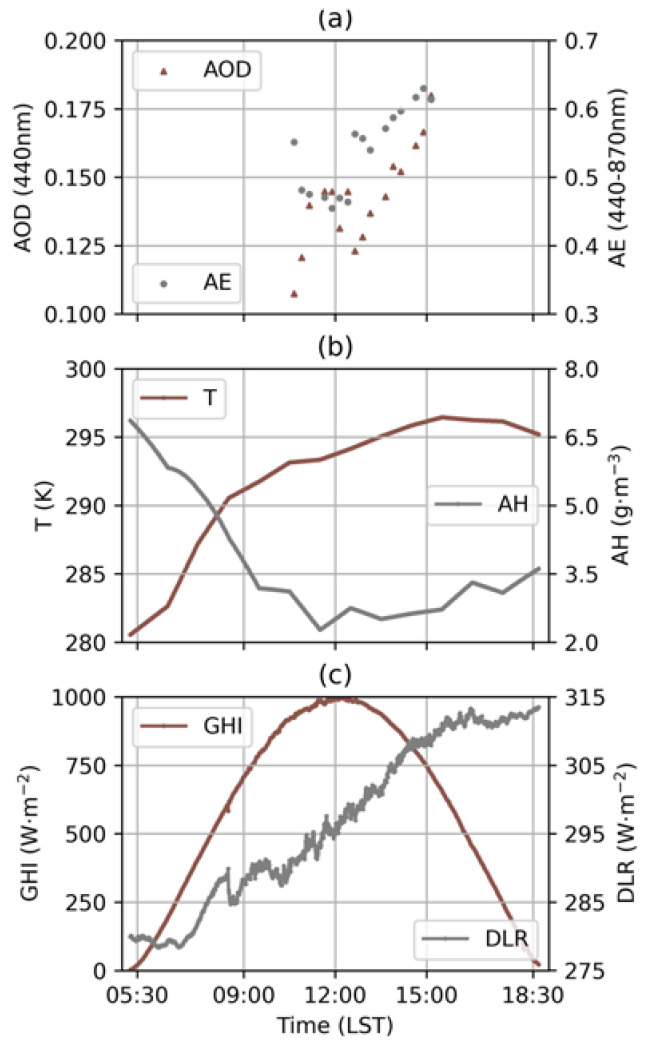
Variation in (**a**) AOD, AE, (**b**) T, AH, (**c**) GHI and DLR during the clean and clear day of 5 May 2018 at Xianghe.

**Figure 5 sensors-24-00397-f005:**
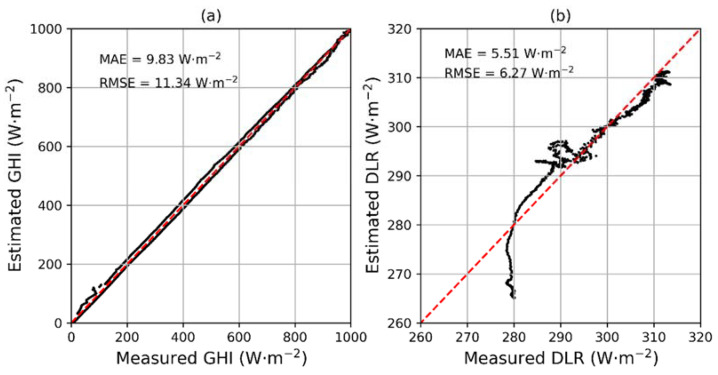
Intercomparison between the measured and the estimated (**a**) GHI and (**b**) DLR for a clean and clear day, i.e., 5 May 2018. The red dashed line is 1:1 line.

**Figure 6 sensors-24-00397-f006:**
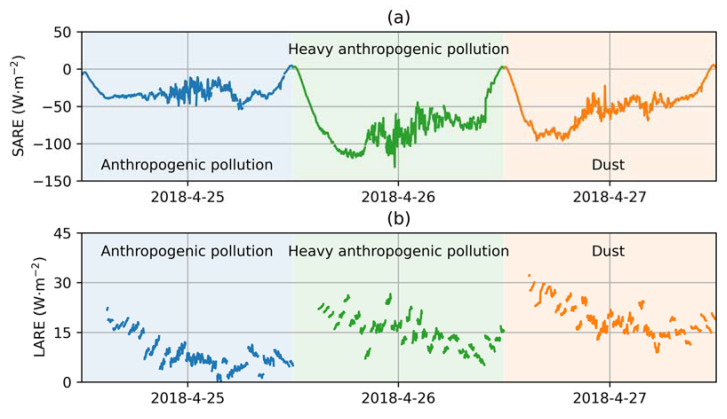
Diurnal variations in (**a**) SARE and (**b**) LARE on 25–27 April 2018 at Xianghe.

**Figure 7 sensors-24-00397-f007:**
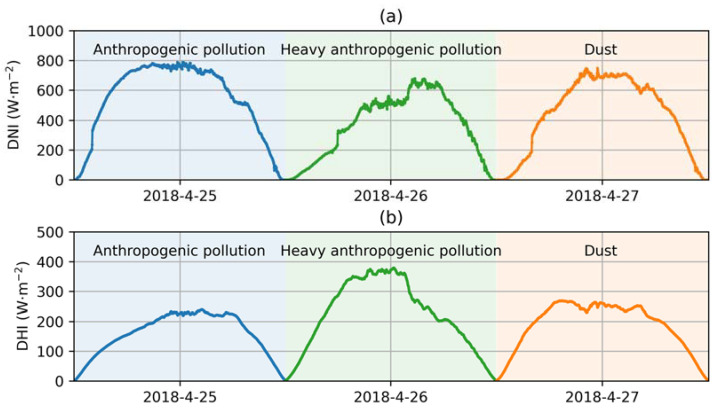
(**a**) DNI and (**b**) DHI at different times on 25–27 April 2018 at Xianghe.

**Figure 8 sensors-24-00397-f008:**
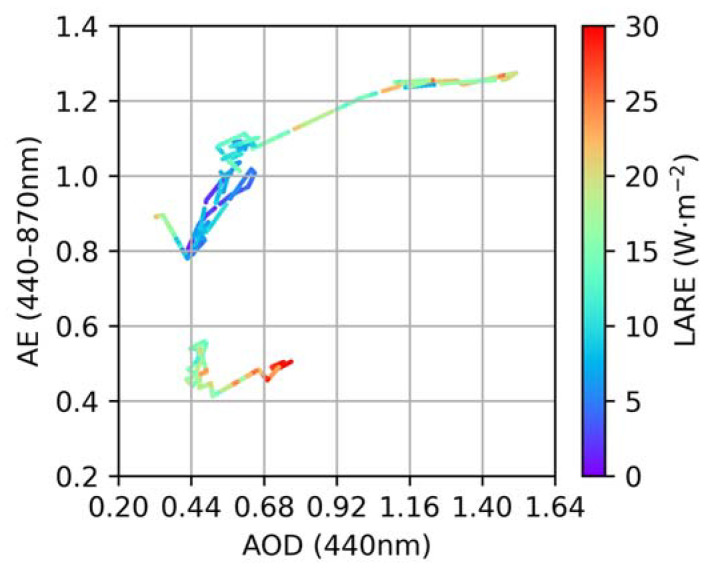
Scattering diagram of the LARE as a function of AOD and AE in the studied episode.

**Table 1 sensors-24-00397-t001:** List of the daily averaged aerosol, atmospheric and radiative properties of the pollution event.

	AOD	AE	T (K)	AH (g/cm^3^)	GHI (W·m^−2^)	DLR (W·m^−2^)
25 April 2018	0.51	0.92	294.98	5.39	534.77	317.02
26 April 2018	0.85	1.14	293.64	7.86	484.06	335.98
27 April 2018	0.5	0.50	293.87	6.01	508.91	333.89

**Table 2 sensors-24-00397-t002:** The daily averaged SARE and LARE during the event.

	SARE (W·m^−2^)	LARE (W·m^−2^)
25 April 2018	−27.57	8.70
26 April 2018	−65.29	15.19
27 April 2018	−51.45	18.99

## Data Availability

Data underlying the results presented in this paper are not publicly available at this time but may be obtained from the authors upon reasonable request.

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
