# Peer review of "Quantitation of the Surface Shortwave and Longwave Radiative Effect of Dust with an Integrated System: A Case Study at Xianghe"

_sensors, 2024, doi:10.3390/s24020397_

Round 1

Reviewer 1 Report

Comments and Suggestions for Authors

General comments:

The radiative effects of dust are crucial for the atmospheric system, and studying the impact of dust on longwave and shortwave radiation holds significant importance. In this paper, the authors conducted a simple computational analysis of a one-day continuous dust event using ground radiation and meteorological observational data. The chosen topic is scientifically meaningful, but some of the conclusions in the article are not entirely convincing. The paper needs improvement in terms of rigor.

Specific comments:

Are the temperature and water vapor data provided in Table 1, Figure 3, and Figure 4 ground-level values? How is the vertical distribution of temperature and water vapor obtained from the microwave radiometer?

Figure 5 shows that the estimation of clear-sky longwave radiation is not accurate during the early morning at sunrise. Therefore, data from this time period should not be considered when calculating the longwave radiation forcing later on. It just so happens that the data in Figure 8 with longwave radiation forcing greater than 30 mostly come from this time period. Such results are not convincing.

The conditions under which Equation 4 can yield reliable computational results need to be discussed in detail.

When presenting the time series, it is suggested to extend the horizontal axis to cover a continuous three-day period and highlight the identified dust events for better visibility.

In addition to surface radiation forcing, the radiation forcing at the top of the atmosphere is equally important. Since the paper proposes to study the radiative effects of dust, it should include the calculation of the radiation forcing at the top of the atmosphere in the title. Additionally, the study of the vertical profile of atmospheric heating rates is also important.

Minor comments / typos:

Figure 1 uses many abbreviations for instruments and variables, and it is necessary to provide specific explanations for the abbreviations when referring to the figure.

Line 120, line 155, and line 166, ‘daily’ should be ‘daily mean’, or ‘daytime mean’ is more appropriate, as nighttime data is not considered.

Author Response

We greatly appreciate the reviewer’s opinions on our submission, please refer to the attachment for detailed reply.

Reviewer 2 Report

Comments and Suggestions for Authors

This paper quantifies the shortwave and longwave radiative effect of a dust using an integrated system. The integrated system includes pyranometers, a solar photometer and a microwave radiometer, which is proposed to study aerosol (especially dust) and its radiative effect. Overall, this article meets the basic requirements for sensors. However, there are a few issues, mostly related to imprecise writing, that detract from the clarity of this article. I suggest that the manuscript can be accepted for publication after some minor revisions.

Minor Comments:

1. Figure 1 appears to be in the wrong position. Please explain the abbreviations in the figure caption, such as e, AH.

2. Page 2, lines 72-75: There are two "pyrgeometers" in the sentence "Specifically, radiation is measured with a pyranometer, pyranometer with shadow ball, pyrgeometer and pyrgeometer; aerosol properties are collected by the sun photometer; meteorological parameters are obtained from a microwave radiometer".

3. In Figure 5, please explain the reason why the estimated DLR is lower than the measured DLR at 280 Wm-2 and consider the impact of this underestimation on the subsequent evaluation of radiation effects.

4. The English of this paper can be improved.

Comments on the Quality of English Language

The English of this paper can be improved.

Author Response

(The authors gave the same response as above.)

Reviewer 3 Report

Comments and Suggestions for Authors

Dear Authors,

Your article "Quantitation of the Shortwave and Longwave Radiative Effect of Dust with an Integrated System: A Case Study at Xianghe" demonstrates the fact that the effects of long-wave radiation disturbance in conditions of high dust aerosol content cannot be neglected in comparison with the disturbance due to the attenuation of short-wave radiation. Careful measurements of shortwave and longwave radiation fluxes along with meteorological parameters during the dust event allowed this conclusion.

I believe that the research was carried out properly and its results are interesting to readers. However, a few annoying typos need to be corrected before publication.

Lines 72-73: The words "pyrgeometer and pyrgeometer" should be replaced with "pyrgeometer and pyrheliometer".

Line 81-83: The sentence beginning with the words “Since 2004” should be rewritten in accordance with the rules of the English language; it has neither a subject nor a predicate.

Line 98: RPG-HATPRO does not measure surface temperature, it measures the vertical profile of air temperature. It is necessary to correct either the word “surface” to the word “air”, or write with what device the surface temperature was actually measured.

Line 109: From my point of view, writing "full day clear sky conditions" is better than writing "a full day clear".

In Figure 3, the temperature axis label says temperature is measured in Celsius degrees, but the values say it is in Kelvin degrees. Please fix one thing.

Line 206: The phrase "the sensitivity between them is conducted" should be replaced with "the sensitivity analysis between them is conducted" or "the sensitivity between them is studied".

Line 216: Please replace "continuously" with "continuous".

In addition, there is an additional optional wish. It would be nice if the authors gave an explanation from the point of view of the physics of the transfer of long-wave radiation of the fact that on the same clear day on April 27 as on April 25 (if you believe Figure 2), the long-wave disturbance of the radiation balance is even higher than it was on April 26. (20.21 > 15.98)  Simply stating this fact is not enough for good study.

Sincerely yours,
Your reviewer.

Comments on the Quality of English Language

Line 81-83: The sentence beginning with the words “Since 2004” should be rewritten in accordance with the rules of the English language; it has neither a subject nor a predicate.

Line 109: From my point of view, writing "full day clear sky conditions" is better than writing "a full day clear".

Line 206: The phrase "the sensitivity between them is conducted" should be replaced with "the sensitivity analysis between them is conducted" or "the sensitivity between them is studied".

Line 216: Please replace "continuously" with "continuous".

Author Response

(The authors gave the same response as above.)

Reviewer 4 Report

Comments and Suggestions for Authors

This study of the shortwave and longwave radiative effects of desert dust over the North China Plain is interesting and brings some information to the reader. It may stimulate future comprehensive research on the point. I have some minor comments which are described below.

1. Abstract. Lines 15 - 16. The actual desert dust event was observed only on April 27, 2018, in accordance with the conditions AOD > 0.4 and AE < 0.6 mentioned in Section 2.2. The presence of desert dust on the other two days (April 25-26) when AE exceeded 0.9 was not proved. Please clarify the presence of desert dust on April 25-26, 2018.

2. Abstract. Line 29. The SARE value of -131.27 W m-2 was observed on April 25, 2018, when the dust presence was not proved. On actual dusty day April 27, SARE peaked at -90 W m-2.

3. Abstract. The shortwave and longwave radiative effects of dust at Xianghe were obtained in comparison to measurements on a clear-sky day using the same monitoring system. It is essential to specify the date with clear-sky conditions in the Abstract and throughout the text.

4. Section 2.1, Line 98. Please clarify what “surface temperature (T)” means. Is it air temperature or skin temperature? At what height temperature (T), absolute humidity (AH), and water vapor pressure (e) were measured?

5. The caption of Table 1. Please replace “daily” by “daily averaged”. The same throughout the text.

6. Figure 3. In this Figure, it is essential to illustrate variations of downward longwave radiation, as well as T and AH not only in the daytime, but also in the nighttime periods: from 0 to 24 local time.

7. Figure 3. It is advisable to add variations of T, AH, GHI, and DLR on the specified clear-sky day to Figure 3, for the convenience of the reader.

8. Figure 4. Similarly to my comments to Fig. 3, in Fir. 4 it is essential to illustrate variations of downward longwave radiation not only in the daytime but also in the nighttime: from 0 to 24 local time.

9. Formula (4). Please specify what “T” stands for. At what height should T be obtained in order to determine downward longwave radiation?

10. The caption of Table 3. Please replace “daily” by “daily averaged”.

11. Figure 7. In this Figure, it is advisable to add variations of DNI and DHI on the specified clear-sky day, for the convenience of the reader.

12. Conclusions. The shortwave and longwave radiative effects of dust were estimated at one location (Xianghe). It is not clear if the obtained results can be extrapolated to the whole area of the North China Plain. Please clarify.

Author Response

(The authors gave the same response as above.)

Round 2

Reviewer 1 Report

Comments and Suggestions for Authors

The DLR parameterization employed in the paper does indeed show the best performance, but the computational errors arising during the early morning cannot be ignored. Although the authors mention that these errors do not affect the assessment of the relative differences in LARE for dust and anthropogenic aerosol, the paper still discusses the absolute magnitudes of LARE in many instances (e.g., Table 3, Figure 6, Figure 8).

In particular, Figure 8 provides key conclusions regarding the size of LARE. To ensure the rigor of the conclusions, it is recommended to exclude data with evident systematic errors in clean DLR estimates during the early morning when conducting statistical analysis. This was also the reason for my previous review comment, asking the authors to confirm the applicability of the DLR parameterization. Despite the authors' belief that the DLR parameterization is applicable at all times, the results from Figure 5 suggest that it is notably ineffective during the early morning.

Furthermore, I noticed significant discrepancies between the values presented in the current version (v2) of Figure 6 and those in the previous version (v1) of Figure 6 (in v2, there are numerous instances of LARE exceeding 30 for the last two days), and these values do not align with those shown in Table 3 and Figure 8. The authors need to conduct a careful examination.

Author Response

Thank you so much for your thorough review. According to your review report, we corrected the manuscript, please check the reply in the attachment and the new version of the manuscript.
